# A Novel Variant in the *TP53* Gene Causing Li–Fraumeni Syndrome

**DOI:** 10.3390/children10071150

**Published:** 2023-06-30

**Authors:** Dimitrios T. Papadimitriou, Constantine A. Stratakis, Antonis Kattamis, Stavros Glentis, Constantine Dimitrakakis, George P. Spyridis, Panagiotis Christopoulos, George Mastorakos, Nikolaos F. Vlahos, Nicoletta Iacovidou

**Affiliations:** 1Endocrine Unit, Second Department of Obstetrics and Gynecology, Aretaieion Hospital, 11528 Athens, Greece; 2Pediatric–Adolescent Endocrinology and Diabetes, Athens Medical Center, 15125 Marousi, Greece; 3Section on Endocrinology & Genetics, The Eunice Kennedy Shriver National Institute of Child Health & Human Development (NICHD), National Institutes of Health (NIH), Bethesda, MD 20814, USA; 4Division of Pediatric Hematology and Oncology, First Department of Pediatrics, National and Kapodistrian University of Athens, 11527 Athens, Greece; 5‘Aghia Sophia’ Children’s Hospital ERN-Genturis Center, 11527 Athina, Greece; 6First Department of Obstetrics and Gynecology, Alexandra University Hospital, National and Kapodistrian University of Athens, 11528 Athens, Greece; 7Surgical Pediatric Oncology, Mitera Children’s Hospital, 15123 Marousi, Greece; 8Pediatric Gynecology Unit, Aretaieion Hospital, 11528 Athens, Greece; 9Second Department of Obstetrics and Gynecology, Aretaieion Hospital, 11528 Athens, Greece; 10Neonatal Department, Aretaieion Hospital, National and Kapodistrian University of Athens, 11528 Athens, Greece

**Keywords:** *TP53* gene, Li–Fraumeni syndrome, adrenocortical tumor, breast cancer, cervical cancer, osteosarcoma

## Abstract

Li–Fraumeni syndrome (LFS) is an autosomal dominant hereditary cancer syndrome associated with germline pathogenic variants in the tumor protein p53 (*TP53*) gene and elevated risk of a broad range of early-onset malignancies. Patients with LFS are at risk of a second and third primary tumor. A 15-month-old girl consulted for clitoromegaly and pubic hair. Adrenal ultrasound detected a large left adrenal tumor. Left total adrenalectomy confirmed adrenocortical carcinoma. Family history revealed multiple highly malignant neoplasms at an early age across five generations, and a genetic dominant trait seemed probable. Whole-genome sequencing was performed. Multiple members of the family were found positive for a novel likely pathogenic variant (c. 892delGinsTTT, p. Glu298PhefsX48, NM_000546.6) in the *TP53* gene, causing the loss of normal protein function through non-sense-mediated mRNA decay. According to the PSV1 supporting criteria and the Auto PVS1 online tool this frameshift variant: hg19/17-7577045-TC-TAAA:NM_000546.6 has a very strong, definitive clinical validity for LFS with autosomal dominant inheritance. Proper guidance resulted in timely diagnosis of a second tumor (primary osteosarcoma) in the index case and in the early detection of breast and cervical cancer in her young mother. Patients with cancer predisposition syndromes like LFS require close multidisciplinary cancer surveillance and appropriate referral to expert centers.

## 1. Introduction

Li–Fraumeni syndrome (LFS) is an autosomal dominant hereditary cancer syndrome associated with germline pathogenic variants in the tumor protein p53 (*TP53*) gene [1] and high risk of a broad range of early-onset malignancies [2]. The majority (70–77%) of LFS-associated tumors are breast cancer, soft-tissue sarcoma, brain tumors, osteosarcoma, and adrenocortical carcinoma [3]. Of note is that close to 50% of children with adrenocortical carcinoma have a *TP53* pathogenic variant. However, ovarian, pancreatic, and gastrointestinal tract tumors are also LFS-related [4]. Patients with LFS are not only at risk of a second and third primary tumor [5] but are also at substantial risk of developing radiation-related second and third cancers [6]. Members of families with Li–Fraumeni syndrome have an exceptionally high risk of developing multiple primary cancers, with the highest risk observed for survivors of childhood cancers [7], the latter needing particularly close monitoring for timely diagnosis of new cancers. In this report, we present how clinical awareness together with detailed family history resulted in the timely diagnosis of an adrenocortical tumor in a child with clitoromegaly, in early detection and radical treatment of breast and then cervical cancer in her young mother, and the subsequent early diagnosis of osteosarcoma in our patient seven years later owing to rigorous follow-up. 

## 2. Case Report

### 2.1. Presentation

A 15-month-old girl (generation V, Figure 1) consulted because of a marked change in clitoris size and pubarche. She had normal blood pressure, no signs of acne, an enlarged clitoris of 1.8 cm, and pubic hair Tanner II. Her bone age maturation was 2 years according to the Atlas of Greulich and Pyle, her length was 83 cm (+1.16 SD, target height −0.15 SD) presenting constant acceleration since the age of 6 months with a height velocity of 22 cm/yr (+5.22 SD), while her BMI had a smooth evolution around +0.5 SD.

### 2.2. Hormonal Profile

Hormonal evaluation at 8:00 h revealed elevated dehydroepiandrosterone sulfate (DHEA-S), Δ4-androstenedione (Δ4), and testosterone, practically 7–10 times higher than the upper normal limit for her age (given in parenthesis): testosterone (T) 1.36 ng/mL (<0.15 ng/mL), Δ4 3.68 ng/mL (<0.5), DHEA-S 145 μg/dL (<15), with a normal corticotropic axis: adrenocorticotropic hormone (ACTH) 27.1 pg/mL, cortisol (F)18 μg/dL), normal serum Νa^+^ and K^+^, with normal vanillylmandelic acid (VMA) and homovanillic acid (HVA) in a random urine spot and only slightly elevated 17-hydroxyprogesterone (17OHP) 1.98 ng/mL (<1.5 ng/mL), practically excluding congenital adrenal hyperplasia (CAH) as a potential cause. Adrenal ultrasound detected a large tumor 5 cm in diameter at the left side, and a low-dose thoracic computerized tomography (CT) scan returned normal.

### 2.3. Surgical Treatment

Open left total adrenalectomy was performed. Histology confirmed adrenocortical carcinoma. The tumor size was 4.8 × 4.5 × 4.6 cm with large cells with eosinophilic and clear cytoplasm and nuclear atypia with multinuclear structures (grade IV according to Fuhrman) with a high mitosis index. Overall, the surgical margins were clear, and the lymph nodes were negative for tumor metastases. The staining and cell markers were vimentin+, melan A+, inhibin+, B-cell leukemia/lymphoma 2 protein (BCL2)+, neuron specific enolase (NSE) + locally, synaptophysin + locally, low-molecular-weight calcium-binding proteins (S100)−, epithelial membrane antigen (EMA)−, carcinoembryonic antigen (CEA)−, chromogranin−, p53+ in 30–35% of the nucleus, and marker of proliferation (Ki67)+ in 15–20%. Her postoperative biology 32 h after surgery showed complete normalization of adrenal androgens: T 0.02 ng/mL, Δ4 0.22 ng/mL, DHEA-S 0.06 μg/dL, and 17OHP 0.83 ng/mL.

### 2.4. Family History—Ordering Whole-Exome Sequencing

Considering the nature of the tumor and the family history (Figure 1): (a) of a maternal cousin aged 15 years who was operated on for an adrenocortical tumor at the age of 5 months and (b) of the maternal grandmother diagnosed of bilateral breast cancer at the age of 35 years, we performed genetic testing. Written informed consent was provided by several family members (#1 our patient, #2 her brother, #3 her father, #4 her mother, #5 her maternal aunt, and #6 her cousin—not the one with the adrenocortical tumor in the past, as her mother refused—and #7 her maternal grandmother), as a genetic dominant trait of maternal descent seemed probable. Our hypothesis was also supported by the fact that other members of the maternal family were diagnosed with osteosarcoma (generation III), tumors of the cervical spine (generation I, II, and III), pancreas (generation III), uterus (generation II), and stomach and colon (generation III), the latter dying at the age of 25 years. Interestingly, three members of the paternal family of our patient’s mother had a history of two highly malignant neoplasms cases: a male subject aged 20 years deceased from lung cancer (generation II) and a female with sarcoma deceased at 24 years (generation III), but no clear transmission pattern to our patient’s mother could be detected (Figure 1). Samples for whole-genome sequencing (WES) and next-generation sequencing (NGS) were collected.

### 2.5. Whole-Exome Sequencing (WES)

In view of the phenotype information, WES analysis in our patient specifically included review of variants in genes associated with adrenal tumors, clitoromegaly, and precocious puberty. No responsible secondary findings were identified in coding regions covered by the XomeDx test for 56 genes, recommended to be reported by the American College of Medical Genetics and Genomics (ACMG) [8].

The index case was found positive (on 22 August 2016) for a novel variant (c. 892delGinsTTT, p. Glu298PhefsX48, NM_000546.6) in the *TP53* gene as a result of a frameshift mutation: 17-7577045-TC-TAAA, which was subsequently found with next-generation sequencing (NGS) in her mother and her maternal grandmother (Table 1). Our patient #1, her mother #4, and her maternal grandmother #7 were positive for the tested variant while her brother #2 and father #3 as well as the maternal aunt #5 and one maternal cousin #6 were assessed and found negative for the variant—all with a free history for malignancies.

The normal sequence with the bases that are deleted in braces and inserted in brackets is: CCAC(G)[TTT]AGCT. The variant detected is predicted to cause the loss of normal protein function through nonsense-mediated mRNA decay according to the AutoPVS1 online tool (https://autopvs1.bgi.com/variant/hg19/17-7577045-TC-TAAA:NM_000546.6, accessed on 25 June 2023) [9]). Considering the current data, the above-described genetic results of targeted carrier testing for this variant and the clinical phenotypes encountered in this family’s members designates this variant in the *TP53* gene as likely pathogenic, in complete accordance with the standards and guidelines for the interpretation of sequence variants [10]. Moreover, according to PSV1 supporting criteria [11], this variant has a very strong, definitive clinical validity for LFS with autosomal dominant inheritance. 

WES was performed only on the child, and subsequent Sanger DNA sequencing of the respective locus was performed in six more family members (Figure 1) by GeneDx (www.geneDx.com, reported on 22 August 2016 for our patient and on 5 May 2017 for the family members). Using genomic DNA from the submitted specimen, the Agilent Clinical Research Exome kit was used to target the exonic regions and flanking splice junctions of the genome. These targeted regions were sequenced simultaneously by massively parallel (NextGen) sequencing on an Illumina HiSeq sequencing system with 100 bp paired end reads. A bi-directional sequence was assembled, aligned to reference gene sequences based on human genome build GRCh37/UCSC hgl9, and analyzed for sequence variants using a custom-developed analysis tool (Xome Analyzer). Capillary sequencing was used to confirm all potentially pathogenic variants identified in this patient. Sequence alterations were reported according to the Human Genome Variation Society (HGVS) nomenclature guidelines. 

### 2.6. Detection of Breast Cancer in the Patient’s Mother

Even before obtaining the results of the WES, our multidisciplinary team clinically examined the mother, aged 26 years, and ordered an extensive laboratory workup including, initially, digital breast tomosynthesis combined with breast ultrasonography and then breast magnetic resonance imaging which revealed unilateral in situ grade III ductal carcinoma of the right breast. She had conventional right breast conservation surgery. The maximum tumor diameter was 2.5 cm. The sentinel lymph node biopsy turned negative with a max lymph node diameter of 0.8 cm. Histology reported a non-invasive ductal carcinoma in situ (DCIS), 90% with extensive myofibroblast stromal reaction, grade III. The staining and cell markers were human epidermal growth factor receptor 2 (HER2)+, cerb-B2 grade III, estrogen receptor 30% weak positive, progesterone receptor <10% positivity, and Ki67 < 10% positivity. Overall, the surgical margins were clear. She received chemotherapy, trastuzumab, and long-term tamoxifen treatment, but one year later, a local invasive tumor in the same site was detected. In view of the incoming positive genetic results, right total mastectomy together with prophylactic total left mastectomy with immediate bilateral reconstruction was offered. The histological findings confirmed local recurrence of the previously mentioned cancer but no signs of dysplasia of the left breast.

### 2.7. Detection of Cervical Cancer in the Patient’s Mother

Five years later, gynecological follow-up showed indices of cervical malignancy. Given the genetic diagnosis and the personal and family history, laparoscopic total hysterectomy with bilateral salpingo-oophorectomy was performed following an SEE FIM protocol (sectioning and extensively examining the fimbriated end of the fallopian tube) with immunochemistry showing positive malignant p53 signatures but negative for Ki67.

### 2.8. Detection of Osteosarcoma in Our Patient

The patient was referred for further follow-up to a center of expertise for cancer predisposition syndromes. Further testing in other family members was performed, and the patient was put under close cancer surveillance. Seven years after the initial diagnosis, a lesion in the tibia was observed on whole-body magnetic resonance imaging (MRI). A biopsy revealed osteosarcoma, and treatment according to the European and American Osteosarcoma Studies (EURAMOS) protocol was initiated. As of the last observation, the patient remains in remission. 

## 3. Discussion

The *TP53* gene encodes a tumor suppressor protein that responds to cellular DNA damage by causing cell cycle arrest, while transcriptionally activating downstream genes to repair the DNA or induce apoptosis (OMIM^®^—Online Mendelian Inheritance in Man^®^ 191170). Heterozygous germline pathogenic variants in *TP53* cause autosomal dominant LFS. LFS is characterized by an increased risk of a broad range of childhood and adult-onset cancers. The following core cancer types account for 70–77% of LFS-associated tumors: breast cancer, soft tissue sarcoma, brain tumors, osteosarcoma, and adrenocortical carcinoma [12,13,14]. Many other cancer types have been reported in association with LFS including ovarian, pancreatic, and gastrointestinal tumors [12]. Individuals with LFS who are diagnosed with cancer have up to a 57% risk of a second primary cancer within 30 years of the first diagnosis and up to a 38% risk of a third primary cancer [7]. Radiation-induced second malignancies have also been reported in individuals with LFS, suggesting that radiation may increase *TP53* pathogenic variant carriers’ risk of subsequent cancers within the radiation field [7,15]. Approximately 24% of LFS cases result from a de novo, rather than inherited, pathogenic variant in the *TP53* gene [16,17].

The likely pathogenic c.892de1GinsTIT variant in the *TP53* gene had not been previously reported to the best of our knowledge at the time that the WES report was issued (22 August 2016). Yet, almost a year later, in June and July 2017, a similar frameshift mutation at this locus (p.E298Cfs*46) has been described in prostate cancers in two studies [18,19]. The c.892delGinsTIT variant causes a frameshift starting with codon glutamic acid 298, changes this amino acid to a phenylalanine residue, and creates a premature stop codon at position 48 of the new reading frame, denoted p.Glu298PhefsX48. This variant is predicted to cause the loss of normal protein function through nonsense-mediated mRNA decay, and according to PSV1 supporting criteria and the report of the AutoPSV1 online tool, the c.892delGinsTTT variant has a very strong, definitive clinical validity for LFS 1 with autosomal dominant inheritance, while it was not observed in approximately 6500 individuals of European and African American ancestry in the NHLBI Exome Sequencing Project, indicating that this variant is not a common benign variant in these populations. While reports on germline variants around this region are found in the literature and can be identified from the *TP53*-IARC database, ClinVar, no specific genotype–phenotype correlation exists. However, based on the evidence presented here, c.892delGinsTTT is a pathogenic variant, related to the adrenal tumor and the osteosarcoma reported in our patient and the breast and cervical cancers in her mother as well as the breast cancer in her maternal grandmother, with clear autosomal dominant inheritance.

The autosomal dominant effect of a heterozygous mutation in the *TP53* gene is particularly interesting. *TP53* is a tumor suppressor gene located on the short arm of chromosome 17. It plays a crucial role in regulating cell division and preventing the formation of tumors, as *TP53* encodes the p53 protein, which acts as a transcription factor and regulates the expression of genes involved in cell cycle arrest, DNA repair, and apoptosis (programmed cell death). A single copy of the mutated *TP53* gene is sufficient to cause an effect, as the loss of function mutations often results in a loss of or reduction in the protein’s normal function, reducing p53 protein’s ability to properly regulate cell division and prevent tumor formation. Therefore, individuals who inherit a single copy of the *TP53* mutation from one parent (heterozygous) can develop the associated phenotype or disease. This dominant negative effect can be possibly explained by mutant p53 proteins interfering with the function of the normal one, the wild-type p53 protein produced from the non-mutated allele. The mutant protein may form non-functional complexes with the wild-type protein, impairing its ability to regulate gene expression and carry out its tumor suppressor functions. Certain *TP53* mutations can also acquire new functions (oncogenic gain of function) or activities that promote tumor development. These gain-of-function mutations may result in altered interactions with other proteins, abnormal gene regulation, or enhanced survival and proliferation of cancer cells [12,20,21,22].

While recent advances in precision medicine show some future promise for feasible use in pediatric oncology [23], this report firstly highlights the importance of analyzing family history. If anything, the diagnosis made in the index case presented, which led through precision medicine to the discovery of this novel deadly variant in the *TP53* gene, and the subsequent timely and hopefully lifesaving interventions in her mother as well as the index case her-self with the early discovery of a new primary osteosarcoma several years later—before any clinical symptoms or signs would appear—could literally have been made to the maternal cousin with the adrenal tumor at the age of 5 months, 15 years ago, meaning the exploitation of a detailed family medical history. Even if some people may present more or less expected or unexpected behaviors expressing denial [24]—as in this girl’s mother who refused genetic testing even when the whole picture was revealed to her by our patient’s mother—our report also raises the issue of the physician’s responsibility to assure the sharing of genetic information with relatives [25], who could possibly benefit if not from preventive interventions at least from early screening, diagnosis, and treatment of an otherwise devastating cancer, as was obviously the case in multiple members through successive generations of this family (Figure 1).

Endocrinology has always been at the forefront of what is today called “precision medicine”, incorporating personalized and genetic data into daily practice [26], playing a critical role in developing and organizing holistic and multidisciplinary approaches for rare diseases in childhood together with precision and personalized oncology [27]. With the increasing availability and lower cost of WES, it has become clear that about 10% of children and adolescents with tumors have germline genetic variants associated with cancer predisposition [28]. Specifically regarding adrenocortical carcinoma, about 50% of affected children present germline mutations in the *TP53* gene, de novo in a quarter of them [29]. Models for integrative genomic analysis of pediatric cancers in clinical practice have been proposed [30], and novel resources for cancer-related genes and potential therapeutic targets in childhood malignancies have been developed [31]. Moreover, the added value of WES beyond a cancer gene panel in selected patients has been proven most valuable in recognizing predisposition in childhood cancer [32].

## 4. Conclusions

Clinical awareness together with detailed family history and a precision medicine approach resulted in timely diagnosis of an adrenocortical tumor in a girl with clitoromegaly and in the early detection of breast and cervical cancer in her young mother and several years later in the detection of a primary osteosarcoma in the same patient in a pediatric oncology reference center. Thus, timely referral of selected patients to organized centers of expertise, like the ones participating in the European Reference Networks, is essential for the care of complex rare diseases like LFS. Thorough genetic analyses, proper and extended genetic counselling including family members with appropriate cancer surveillance protocols are critical for optimal outcomes in cases of cancer predisposition syndromes. 

## Figures and Tables

**Figure 1 children-10-01150-f001:**
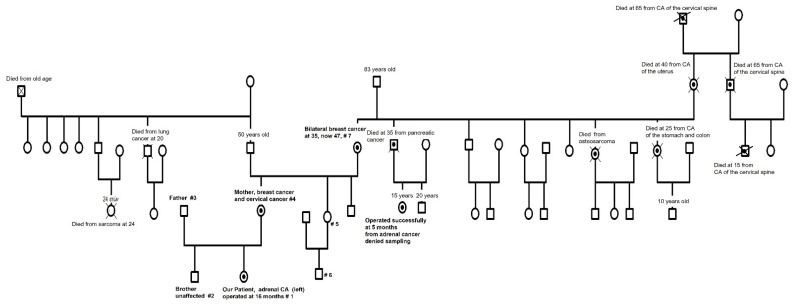
Family tree. #1–7: subjects genetically tested, ●: subjects found or presumed to be carrying one copy of the new *TP53* variant.

**Table 1 children-10-01150-t001:** Results of targeted whole exome sequencing: causative variants in the disease genes associated with the reported phenotype.

Gene	Disease	Mode of Inheritance	Variant	Coding DNA	Zygosity	Inherited from	Classification
TP53	Li–Fraumeni Syndrome	Autosomal Dominant	p.E298FfsX48	c.892delGinsTIT	Heterozygous	Proband’s mother and maternal grandmother	Likely Pathogenic Variant

## Data Availability

The variant information can be found at: https://autopvs1.bgi.com/variant/hg19/17-7577045-TC-TAAA:NM_000546.6 (accessed on 25 June 2023).

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
