# Peer review of "A Novel Variant in the TP53 Gene Causing Li–Fraumeni Syndrome"

_children, 2023, doi:10.3390/children10071150_

Round 1
Reviewer 1 Report
Dear authors, thank you for your article. I find it interesting, but I believe requires some changes.
Abstract, line 34 - please, do not use devastating cancers. This is a scientific paper, there are better expressions.
Abstract, line 36 - the reported variant is likely pathogenic, please, stick to this classification throughout your whole paper. In one paragraph you state it is pathogenic, at the end of the paragraph you state it is likely pathogenic. It should be the same in your whole paper.
Case report, line 71, 76 - please, write the whole words, before using the abbreviation. Some of the readers may not be familiar with these abbreviations. It applies to the whole paper, these are just two examples.
Case report, line 102 - the same comment about devastating. Do you mean deceased instead of diseased?
Figure 1 - please, correct this figure, Your pedigree tree is wrong. We indicate the proband with an arrow and all of the affected individuals are in dark color. The symbol you have used indicates that they are carriers of an autosomal recessive allele, Li-Fraumeni is an autosomal-dominant condition. Indicate the different generations and number the individuals from the specific generation. There is no need to write X: diseased, this is well-known.
WES, line 113 - the right term is pathogenic variant, not pathologic. Please, add the SNP ID from Clinvar, I could not find it in the database. Was the information uploaded in the Clinvar?
Detection of breast cancer in the patient’s mother, line 118 - please define extensive laboratory workup. You cannot detect cancer based only on lab tests.
Materials and method - should be merged with WES from your case-report, the structure of the paper is very confusing.
Materials and method, lines 156-164 - very detailed explanation, there is no need, this is something everyone could check for themselves
Results - should be merged with case description, your paper could have the following structure: Introduction, case report, discussion, conclusion.
Results, line 183 - is the variant pathogenic or likely pathogenic, there is a difference
Table 1 - from your paper is seem like the variant was inherited from patient's grandmother, not the mother
Discussion, lines 214 - 226 - this is your personal view on the subject of reporting about the likely pathogenic variant to other members of the family. This is not allowed, this is personal medical information and we cannot share it with other family members, even if they could be in risk of hereditary cancer. There are very strict guidelines on this topic and that the patient is the only one who could share the news with other family members. Stick to the guideline for genetic counseling, not to your personal beliefs.
Discussion, lines 227 - 233 - you repeat information from the previous paragraph, there is no need
Discussion, lines 234, 236 - endocrinology is not solemnly responsible for “precision medicine”, nor is pediatric endocrinology the only specialty involved in managing patients with rare diseases. These patients should be referred to expert centres.
The eleven authors of this case report strike me as being really odd. Although I recognize that this is a team work, 11 persons for a case report seems a bit excessive considering that this is not a clinical trial. However, the choice of which authors to list is entirely up to you.
There are several mistakes, the text should be checked again.
Author Response
Response to R1
Dear authors, thank you for your article. I find it interesting, but I believe requires some changes.
A: We thank you for your constructive remarks
Abstract, line 34 - please, do not use devastating cancers. This is a scientific paper, there are better expressions.
A: We changed this expression to “highly malignant neoplasms”
Abstract, line 36 - the reported variant is likely pathogenic, please, stick to this classification throughout your whole paper. In one paragraph you state it is pathogenic, at the end of the paragraph you state it is likely pathogenic. It should be the same in your whole paper.
A: We kept “likely pathogenic” according to your proposal throughout the manuscript although we now provide clear evidence that this variant is definitely pathogenic with PVS1 supporting criteria and according to the AutoPVS1 online tool report (https://autopvs1.bgi.com/variant/hg19/17-7577045-TC-TAAA:NM_000546.6)
Case report, line 71, 76 - please, write the whole words, before using the abbreviation. Some of the readers may not be familiar with these abbreviations. It applies to the whole paper, these are just two examples.
A: We analysed all acronyms throughout the manuscript except DNA/RNA
Case report, line 102 - the same comment about devastating. Do you mean deceased instead of diseased?
A: we changed also to “highly malignant” and corrected “diseased” to “deceased”
Figure 1 - please, correct this figure, Your pedigree tree is wrong. We indicate the proband with an arrow and all of the affected individuals are in dark color. The symbol you have used indicates that they are carriers of an autosomal recessive allele, Li-Fraumeni is an autosomal-dominant condition. Indicate the different generations and number the individuals from the specific generation. There is no need to write X: diseased, this is well-known.
A: We clarified now in the figure legend that “●: subjects found or presumed carrying one copy of the new TP53 variant” and we omitted “X: diseased”. Although most pedigrees presented in the literature follow the pattern you correctly propose, homozygous Li Fraumeni syndrome exists, and refers to a rare scenario where an individual inherits two mutated copies of the TP53 gene, one from each parent. As in: Evans DG, Wu CL, Birch JMBRCA2: a cause of Li–Fraumeni-like syndrome. Journal of Medical Genetics 2008; 45:62-63, a Li Fraumeni pedigree can be also presented with marked heterozygosity. Since in our pedigree there was only one copy affected, we chose this mark (not putting in full black the affected individuals), but not half-blacked either (as there is a dominant effect) to show that affected individuals are heterozygotes.
Arrows were added as proposed.
WES, line 113 - the right term is pathogenic variant, not pathologic. Please, add the SNP ID from Clinvar, I could not find it in the database. Was the information uploaded in the Clinvar?
A: we corrected “pathologic” to “pathogenic”
We thought it was by GeneDx, but it appears that you are right and that it has not. We initiated the ClinVar Organization submission: SUB13570840 and we shall complete the task. ClinVar number will be probably included in the proofs – revision awaited. We also updated the manuscript with the newest version of the NCBI Reference Sequence: NM_000546.6 that replaced NM_000546.4
Detection of breast cancer in the patient’s mother, line 118 - please define extensive laboratory workup. You cannot detect cancer based only on lab tests.
A: we now specified as requested “including initially digital breast tomosynthesis combined to breast ultrasonography and then breast magnetic resonance imaging”
Materials and method - should be merged with WES from your case-report, the structure of the paper is very confusing.
A: We are sorry if the article’s outline gives that impression. We merged sections as suggested and extensively revised the whole paper
Materials and method, lines 156-164 - very detailed explanation, there is no need, this is something everyone could check for themselves
A: We agree and lines 156-164 were deleted.
Results - should be merged with case description, your paper could have the following structure: Introduction, case report, discussion, conclusion.
A: We merged sections as suggested.
Results, line 183 - is the variant pathogenic or likely pathogenic, there is a difference
A: We have revised that throughout the manuscript as you previously suggested
Table 1 - from your paper is seem like the variant was inherited from patient's grandmother, not the mother
A: We clarified, since they both had the mutation and Li Fraumeni syndrome: “Proband’s mother and maternal grandmother”
Discussion, lines 214 - 226 - this is your personal view on the subject of reporting about the likely pathogenic variant to other members of the family. This is not allowed, this is personal medical information and we cannot share it with other family members, even if they could be in risk of hereditary cancer. There are very strict guidelines on this topic and that the patient is the only one who could share the news with other family members. Stick to the guideline for genetic counseling, not to your personal beliefs.
A: We think that maybe you have misunderstood this part of the discussion, and that is why we now clarify by adding the explanatory phrase in yellow: “While recent advances in precision medicine show some future promise for feasible use in pediatric oncology [16], this report firstly highlights the importance of analyzing family history. If anything, the diagnosis made in the index case presented, which led through precision medicine to the discovery of this novel deadly variant in the TP53 gene, and the subsequent timely and hopefully lifesaving interventions in her mother as well as the index case her-self with the early discovery of a new primary osteosarcoma several years later - before any clinical symptoms or signs would appear - could literally had been made to the maternal cousin with the adrenal tumor at the age of 5 months, 15 years ago, meaning the exploitation of a detailed family medical history.”
It is possible that at that time the treating physicians did not pay the necessary attention to that patient’s family history, or that this mother did not provide any information if asked. We cannot know and that patient was not treated by any member of this team.
In our case-family, the patient’s mother and only her - who also turned a patient with LFS - informed her relatives.
The mother of our proband’s cousin-girl operated also from adrenocortical carcinoma 15 years before our patient’s was operated, refused genetic testing to her-self and her daughter also depriving her-self and her daughter from receiving precautionary medical advice and explorations – although this was offered with no charge. This attitude is discussed with reference #17 and the issues that are possibly raised concerning our responsibility as physicians with reference #18. Our opinion is that there is an issue, we have the obligation to discuss it, and we do it very carefully; then again, we never breached patient confidentiality nor the existing guidelines at any timepoint.
Discussion, lines 227 - 233 - you repeat information from the previous paragraph, there is no need
A: We omitted repeated information as proposed and added a Conclusion section as proposed above
Discussion, lines 234, 236 - endocrinology is not solemnly responsible for “precision medicine”, nor is pediatric endocrinology the only specialty involved in managing patients with rare diseases. These patients should be referred to expert centres.
A: We agree, transformed the phrase, and added a relevant reference: “Endocrinology has always been at the forefront of what is called today “precision medicine” incorporating personalized and genetic data in daily practice [23], playing a critical role in developing and organising holistic and multidsciplinary approaches for rare diseases in childhood together with precision and personalised oncology [24].
The eleven authors of this case report strike me as being really odd. Although I recognize that this is a team work, 11 persons for a case report seems a bit excessive considering that this is not a clinical trial. However, the choice of which authors to list is entirely up to you.
A: Respecting your reservation we asked the only author, resident in Endocrinology in Switzerland at this time, Christina Bothou, that was not clinically involved in the multidisciplinary team and in patient treatment or follow-up but only responsible for data gathering and acquisition, to withdraw from authorship, and she agreed.
Reviewer 2 Report
In this case report, the authors identify a TP53 mutation in a young female patient which is likely to have been the cause of an adrenal tumor. The study is of some significance, but it lacks integration with current literature. I provide some point of improvement below.
1. The TP53 mutation identified at this locus is not as novel as the author make it sound. A similar frameshift mutation at this locus (p.E298Cfs*46 )has been described in prostate cancers in two previous studies (PubMed IDs: 28481359, 28825054). This mutation is not unlike the ones that the authors identified here which produces a frameshift and probable inactivation of protein function.
2. The authors assert that “The variant detected is predicted to cause loss of normal protein function either through protein truncation or non-sense mediated mRNA decay.” This is a guess and not necessarily a conclusion. The authors used the Richards et al guidelines, but these recommendations are somewhat dated. We now have artificial intelligence algorithms that will classify mutations as pathogenic, driver, or passenger. Rather than guessing the effect on protein function, the authors should use these established algorithms. I suggest, OPEN CRAVAt or CBioPortal. Even Richards et al lists some algorithms that could be used to determine the effect on the function (See table 2 in their paper).
3. Other TP53 frameshift truncations have been identified in this region. Please check COSMIC for previously identified mutations. These types of mutations affect the tetramerization domain of TP53 or nuclear localization signal. How do the authors’ identified mutation compare with previously identified truncations in the same TP53 region.
4. Finally, the authors conclude that their identified mutation is pathogenic, yet it occurs in a heterozygous context. Since TP53 is a tumor suppressor gene, how do the authors explain that a heterozygous mutation can be pathogenic? A discussion should be introduced to explain this observation. Recent analysis of human cancer genomes (e.g. TCGA or COSMIC, etc) have shown that heterozygocity of even bona fide tumor suppressor genes is sufficient to produce a cancer phenotype but the authors make no mention of haploinsufficiency. TP53 and other checkpoint and DNA damage repair genes have been front and center in understanding this phenomena. The authors should dedicate a paragraph in the discussion on haploinsufficiency.
English is OK.
Author Response
In this case report, the authors identify a TP53 mutation in a young female patient which is likely to have been the cause of an adrenal tumor. The study is of some significance, but it lacks integration with current literature. I provide some point of improvement below.
A: We thank you for your constructive remarks
- The TP53 mutation identified at this locus is not as novel as the author make it sound. A similar frameshift mutation at this locus (p.E298Cfs*46 )has been described in prostate cancers in two previous studies (PubMed IDs: 28481359, 28825054). This mutation is not unlike the ones that the authors identified here which produces a frameshift and probable inactivation of protein function.
A: Thank you! We incorporated this in the discussion as follows: “The likely pathogenic c.892de1GinsTIT variant in the TP53 gene had not been previously reported to the best of our knowledge at the time of the WES report was issued (22 August 2016). Yet, almost a year later, in June and July 2017, a similar frameshift mutation at this locus (p.E298Cfs*46) has been described in prostate cancers in two studies [16,17].”
- The authors assert that “The variant detected is predicted to cause loss of normal protein function either through protein truncation or non-sense mediated mRNA decay.” This is a guess and not necessarily a conclusion. The authors used the Richards et al guidelines, but these recommendations are somewhat dated. We now have artificial intelligence algorithms that will classify mutations as pathogenic, driver, or passenger. Rather than guessing the effect on protein function, the authors should use these established algorithms. I suggest, OPEN CRAVAt or CBioPortal. Even Richards et al lists some algorithms that could be used to determine the effect on the function (See table 2 in their paper).
Α: For germline variants Richards et al guidelines are still the official publication when referring to pathogenicity. Unfortunately, CBioPortal and CRAVAT are mainly for somatic mutations. According to the ACMG/AMP guidelines our variant is classified as likely pathogenic, with PVS1 supporting criteria and the PVS1 flowchart derived from the AutoPVS1 tool: https://autopvs1.bgi.com/variant/hg19/17-7577045-TC-TAAA:NM_000546.6
Section “2.4 Whole Exome Sequencing (WES)” has now been completely revised with all the above information and the relevant references.
- Other TP53 frameshift truncations have been identified in this region. Please check COSMIC for previously identified mutations. These types of mutations affect the tetramerization domain of TP53 or nuclear localization signal. How do the authors’ identified mutation compare with previously identified truncations in the same TP53 region.
A: There are reports on germline variants around this region in the literature, which we can identify from the TP53-IARC database and ClinVar (Cosmic database reports somatic mutations) but no specific genotype-phenotype correlation exists. We now added this information to the manuscript and discuss it L215-227. Pathogenic variants that affect the tetramerization of the protein are missense gain-of-function mutations with a different pathogenic mechanism.
- Finally, the authors conclude that their identified mutation is pathogenic, yet it occurs in a heterozygous context. Since TP53 is a tumor suppressor gene, how do the authors explain that a heterozygous mutation can be pathogenic? A discussion should be introduced to explain this observation. Recent analysis of human cancer genomes (e.g. TCGA or COSMIC, etc) have shown that heterozygocity of even bona fide tumor suppressor genes is sufficient to produce a cancer phenotype but the authors make no mention of haploinsufficiency. TP53 and other checkpoint and DNA damage repair genes have been front and center in understanding this phenomena. The authors should dedicate a paragraph in the discussion on haploinsufficiency.
A: Thank you for this suggestion. We now added a relevant paragraph in the discussion as suggested L229-246.
Round 2
Reviewer 1 Report
Thank you for changing the manuscript according to my suggestions.
Reviewer 2 Report
The authors have either made the changes I recommended or provided compelling arguments. This reviewer is satisfied and recommends publication.